



# Spectral Characterization, Radiative Forcing, and Pigment Content of Coastal Antarctic Snow Algae: Approaches to Spectrally Discriminate Red and Green Communities and Their Impact on Snowmelt

Alia L. Khan[1,2], Heidi Dierssen[3], Ted Scambos[4], Juan Höfer[5,6], Raul R. Cordero[7]

[1]Department of Environmental Sciences, Huxley College of the Environment, Western Washington University, Bellingham, WA, USA

[2]National Snow and Ice Data Center, Cooperative Institute for Research in Environmental Sciences, University of Colorado – Boulder, Boulder, CO, USA

[3]Department of Marine Sciences and Geography, University of Connecticut, Groton, CT. USA

[4]Earth Science and Observation Center, Cooperative Institute for Research in Environmental Sciences, University of Colorado – Boulder, Boulder, CO, USA

[5]Escuela de Ciencias del Mar, Pontificia Universidad Católica de Valparaíso, Valparaíso, Chile.

[6]Centro FONDAP de Investigación en Dinámica de Ecosistemas Marinos de Altas Latitudes (IDEAL), Valdivia, Chile

[7]Department of Physics, University of Santiago, Av. Bernardo O'Higgins 3363, Santiago, Chile

*Correspondence to*: Alia L. Khan (alia.khan@wwu.edu)

**Abstract.** Here, we present radiative forcing (RF) estimates by snow algae in the Antarctic Peninsula (AP) region from multi-year measurements of solar radiation and ground-based hyperspectral characterization of red and green snow algae collected during a brief field expedition in austral summer 2018. Our analysis includes pigment content from samples at three bloom

sites. Algal biomass in the snow and albedo reduction are well-correlated across the visible spectrum. Relative to clean snow, visibly green-patches reduce snow albedo by ~40% and red-patches by ~20%. However, red communities absorb considerably more light per mg of pigment compared to green communities, particularly in green wavelengths. Based on our study results, it should be possible to differentiate red and green algae using Sentinel-2 bands in blue, green and red wavelengths. Instantaneous RF averages were double for green (180 W m$^{-2}$) vs. red communities (88 W m$^{-2}$), with a maximum of 228 W m$^{-2}$.

$^{2}$. Based on multi-year solar radiation measurements at Palmer Station, this translated to a mean daily RF of ~26 W m$^{-2}$ (green) and ~13 W m$^{-2}$ (red) during peak growing season – on par with mid-latitude dust attributions capable of advancing snowmelt. This results in ~2522 m$^{3}$ of snow melted by green-colored-algae and ~1218 m$^{3}$ of snow melted by red-colored-algae annually over the summer, suggesting snow algae play a significant role in snowmelt in the AP regions where they occur. We suggest impacts of RF by snow algae on snowmelt be accounted for in future estimates of Antarctic ice-free expansion in the AP

region.



## 1 Introduction

Snow algae blooms are common in coastal snow packs of the northern Antarctic Peninsula (AP) and adjacent islands during austral summer. Over the past 50 years, the AP has warmed significantly (Turner et al., 2005) and although a recent slowing of this trend has been observed due to natural variability (Turner et al., 2016), it remains one of the most rapidly warming regions on the planet (Hansen et al., 2010; Steig et al., 2009; Turner et al., 2009, 2014; Vaughan et al., 2003). This intense warming is likely increasing snowmelt availability, potentially impacting red and green snow algae blooms, which are sensitive

to light (Rivas et al., 2016), water and temperature (Hoham and Remias, 2020). Significant changes have been identified in the regional sensitivity of moss growth to past temperature rises, suggesting Antarctic terrestrial ecosystems will alter rapidly under future warming (Amesbury et al., 2017; Gutt et al., 2017). An 'Antarctic greening' is therefore possible, following well-documented Arctic observations i.e., (Myers-Smith et al., 2020).

Snow packs serve as habitats and nutrient reservoirs for microbes across the global cryosphere, e.g., (Hisakawa et al.,
2015; Hodson et al., 2017; Lutz et al., 2016; Remias et al., 2005; Segawa et al., 2018). Meltwater can stimulate a return to an active growing state through the ionic pulse, when there is a flushing and mass-loading of ions and nutrients at the onset of snowmelt, e.g., (Williams and Melack, 1991). In coastal Antarctic snowmelt, nutrient cycling is influenced by inputs from penguin and seal excreta, which helps to fertilize local glacial, terrestrial, and aquatic ecosystems (Hodson, 2006). The input of snowmelt into coastal regions could have significant impacts on marine primary production, such as during high melt years
which has been shown to increase primary production in the ocean (Dierssen et al., 2002).

Light absorbing particles (LAPs) (Skiles et al., 2018) are generally comprised of dust e.g., (Bryant et al., 2013; Painter et al., 2012; Skiles and Painter, 2019), black carbon e.g.,(Khan et al., 2019; Rowe et al., 2019), volcanic ashes (Flanner et al., 2007) and snow algae e.g.,(Ganey et al., 2017; Lutz et al., 2016). LAPs influence spectral albedo in the visible spectrum, 400 – 700 nm (Warren and Wiscombe, 1980). Spectral albedo is further dependent on physical snow properties such as specific
surface area, i.e., grain size and shape, e.g.,(Cordero et al., 2014), liquid water content, surface roughness, snow depth, albedo of underlying ground (for thin snow packs), and snow density (Flanner et al., 2007). Aged snow that has collected LAPs, typically has an albedo around $0.5 – 0.7$, and in extreme cases of organic LAP content, can range below $0.2$ (Khan et al., 2017). Arctic red snow algae blooms can reduce surface albedo up to 13% (Lutz et al., 2016). Snow darkening by LAPs and the associated radiative forcing (RF) have the potential to impact the long-term climate, while accelerating snow melt and changes
in regional hydrology in the near term. Given their anthropogenic association they are not expected to diminish in the future (Skiles et al., 2018). However, unlike other LAPs (Flanner et al., 2007), the 'bioalbedo' feedback (Cook et al., 2017) from microbes living and growing on the surface of the cryosphere is not currently accounted for in global climate models and only one study has assessed the RF of red snow algae, in Alaska (Ganey et al., 2017).





In our study region, the snow algae are comprised of green algae (Chlorophyta) and based on findings on nearby
Adelaide Island, are likely a combination of *Chloromonas*, *Chlamydomonas*, and *Chlorella* genuses (Davey et al., 2019). A
discrepancy in the literature appears to exist as to whether the algae that appear green vs red are from the same species and
reflect a progression of the snow algae life-cycle or are two distinct algal species. For example, a previous study of snow algae
on King George Island suggested the red and green snow algae are from the same species based on pigment and fatty acid
analysis (Kim et al., 2018). However, Davey et al. (2019) showed that green communities contain higher chlorophyll
concentrations than red communities and were comprised of *Chloromonas, Chlamydomonas,* and *Chlorella,* whereas
*Chloromonas* was the only genus of snow algae identified in red blooms. Hoham and Remias (2020) state the pigment
composition can vary depending on the stage of the life-cycle of chlamydomonadalean snow algae (e.g. *Chloromonas*,
*Chlamydomonas*, and *Chlainomonas)* and although the reproductive stages are distinct, they remain unclear and relatively few
studies have documented snow algae on Antarctica since 2000 (Davey et al., 2019a; Fujii et al., 2010; Ling, 2001; Procházková
et al., 2019; Remias et al., 2013b, 2013a). Multiple studies have also found that when both red and green snow algae were
present, a complex community of bacteria, protists and fungi are generally also present (Davey et al., 2019; Hodson et al.,
2017). In this study, we did not have the opportunity to analyze the community composition of our samples, but plan to include
this in future work.

Recent developments in multi-spectral satellite-based instrumentation have advanced our ability to monitor and map
Antarctica's terrestrial biosphere (Fretwell et al., 2011). Algorithms for retrieval of aerosol and snow/ice properties in polar
and mid-latitude regions are actively developing, e.g. (Stamnes et al., 2007). However, detection of biological constituents
remains challenging due to the presence of other LAPs, such as dust (Huovinen et al., 2018). Snow algae was first mapped in
the Sierras using the chlorophyll absorption feature at 680 nm (Painter et al., 2001). Red/green band ratios have since been
applied to SPOT (Takeuchi et al., 2006) and Landsat-8 imagery (Ganey et al., 2017; Hisakawa et al., 2015), which appears to
work well in regions with limited exposure of bare rock and soil. On Greenland, Wang et al. (2018) used Sentinel-3 imagery
to map the spatial pattern of glacier algae using the reflectance ratios between 709 nm and 673 nm. Within the AP region,
spectral mixture analysis with Sentinel 2 imagery can more reliably classify snow algae when other LAPs are present
(Huovinen et al., 2018) and multi-year mosaics have identified green-snow as a terrestrial carbon sink (Gray et al., 2020). We
build on these pioneering studies by presenting new approaches to spectrally distinguish red from green snow algae and further
evaluate the RF of snow algae based on environmental conditions in this part of Antarctica.

Snow algae can be used to monitor climate impacts since their habitat is highly responsive to environmental conditions
(Hoham and Remias, 2020). Here we present ground-based observations of surface spectral albedo and corresponding algal
biomass from three coastal snow-sites in the South Shetland Islands of Antarctica at the height of the austral 2018 growing
season. In this study we targeted patches of snow that are clean (free of snow algae), visibly red, visibly green, and mixed
(visibly red and green) in order to calculate RF and develop algorithms for eventual remote sensing of snow algae in order to
monitor their response to climactic change (Figure 1). The following analysis considers the reflectance measurements in terms
of albedo, RF and the variability in reflectance in relationship to pigment concentration and light absorption. Finally, we



evaluate several heritage approaches to estimate pigment concentration that can be applicable to a variety of high spatial resolution satellites, including Sentinel-2.

## 2.0 Methods

### 2.1 Site Description and Site Selection

Field observations were conducted at two sites on King George Island (KGI), the largest of the South Shetland Islands, and one site on northern Nelson Island (NI), southwest of KGI in January 2018. The first site on KGI was located near Fildes/Maxwell Bay between the Chilean Prof. Julio Escudero Station and the Chinese Great Wall Station. It is approximately 200 meters above mean high tide, with slightly less frequent wildlife traffic. The second KGI site was located in Collins Bay adjacent to Collins Glacier and is approximately 100 meters above mean high tide, with more frequent seal, penguin and other bird activity. Both KGI sites were flat, low-sloping south-east facing beaches (Figure 1). The third site on Nelson Island is a low-sloping north-west facing beach at the edge of Nelson Glacier with frequent seal and penguin activity due to the location approximately 25 meters above mean high tide. The weather conditions were uniformly cloudy at Fildes and Nelson and clear at Collins Bay.

Optically thick (> 30cm) snow packs were prioritized for spectral albedo data acquisition and corresponding snow algae sampling in order to minimize the impact of the underlying ground on spectral albedo. Sites were also selected based on where it was possible to sample 1) a control site with relatively clean snow having no visible snow algae 2) green snow algae, 3) red snow algae and 4) mixed-phase green and red algae. At each site, duplicates of each snow type were measured with the spectrometer (except at Nelson Island where only one Mixed site was observed). All samples were collected around noon local Chilean time, when the seasonal snow pack was also receiving the most incoming solar radiation. As evidenced in Figure 1, this can also result in snowmelt ponding. While the snowpack contained a lot of water, large areas of melt ponding were avoided for spectral measurements. Snow depth was measured at each observation site and reported to the nearest centimeter if less than one meter (Table 1).

### 2.2 Algal Pigment Concentration

Surface snow samples (top 10 cm) were collected at each site in plastic bags, melted at room temperature and filtered through 0.45 µm pore size nucleopore filters. The filters were immediately frozen (−20 °C) and then shipped frozen to a laboratory in Valdivia, Chile where they were fluorometrically analyzed (Turner Design TD-700), using 90% acetone for pigment extraction for Chlorophyll-*a* and phaeopigment analysis according to standard procedures (Parsons et al., 1984). Pigment was considered to be the sum of Chlorophyll-*a* determined fluorometrically and the amount attributed to phaeopigments (ChP). The amount of phaeopigment in the samples was quite high (2.34 times higher than Chlorophyll *a*), but was well-correlated to chlorophyll *a* concentration (Fig. 2, $r^2$=0.97, m=2.34).



Phaeopigments are a degradation product of algal chlorophyll pigments and the high amounts could be related to a combination of environmental and methodological factors. The cold temperatures of the Antarctic may allow for pigment degradation products of algae to remain in the snow longer than in other types of environments. Secondly, snow algae are known to have high amounts of Chlorophyll-*b* at levels up about half the amount of Chlorophyll-*a* (Davey et al., 2019a). Fluorometric methods can underestimate Chlorophyll-*a* and substantially overestimate phaeopigments in phytoplankton when Chlorophyll-*b* is present (Vernet and Lorenzen, 1987). Finally, higher than normal phaeopigments can also occur due to issues in shipping and storing samples over time. For all of these reasons, we have chosen to add the amount of Chlorophyll-*a* and phaeopigments together in this paper and refer to this as ChP. Davey et al., (2019, Table 1) reported the dry cell mass (DCM) pigment composition of snow algae communities on nearby Adelaide Island with green communities mostly comprised of Chlorophyll-*a* (42%), followed by Chlorophyll-*b* (19%), Lutein (7%), ß-Carotene (5%), and Astaxanthin-esters (4%). Red communities were mostly comprised of Astaxanthin-esters and Astaxanthin-like_esthers (60%), Chlorophyll-*a* (24%), Chlorophyll-*b* (11%), Lutein (< 1%), and Xanthophyll (5%).

Pigment measurements were calculated in volumetric units of melted snow and hence vary with the depth and volume of snow excavated from the site. Following Painter et al. (2001) and Thomas (1972), the top 10 cm were selected for this analysis. However, the distribution of algae with depth in the snow is not well documented and the pigment measurements will be more or less dilute depending on the excavation depth. Hence it cannot be compared across regions. We are unaware of a community standard for determining excavation depths and published studies have used depths varying from 5 - 10 cm (Davey et al., 2019a; Grinde, 1983) and the algae may be covered by varying depths of snow, which could be several meters thick although scouring from strong blizzards may later expose sites and facilitate dispersal of the cells (Ling and Seppelt, 1993). In order to rectify this and make our data relevant to other studies and future work, we standardized these values to represent the amount of integrated pigment per surface area of snow (mg ChP m$^{-2}$). This more standardized metric is also considered more appropriate for conducting comparisons to snow albedo measured above the snow surface. We also recommend researchers excavate appropriately deep in the snow to harvest the majority of the algae with depth. For this conversion, we calculated area-weighted ChP following (Eq. 1) using a mean snow density of 610 kg m$^{-3}$ and excavated snow depth was generally 10 cm (0.10 m), except for select samples at Nelson that were 7-8 cm (0.07 m). In the formulation, "dry" refers to native snow and "melt" refers to melted snow, following Eq. (1):

$$\frac{mg\ ChP}{m^2(dry)} = \frac{ug\ ChP}{L} \times \frac{1000\ L}{m^3(melt)} \times \frac{mg}{1000\ ug} \times \frac{m^3(melt)}{1000\ kg} \times \frac{610\ kg}{m^3(dry)} \times \frac{Depth\ m(dry)}{1} \qquad (1)$$

This parameter could underestimate the amount of integrated pigment, if there was significant algae beneath excavated snow layer. However, the 10 cm excavation depth selected in this study was the depth at which color was no longer present in the samples and hence most of the algae was believed to be harvested within the top layer.



## 2.2 Ground Based Spectral Albedo Measurements

Spectral reflectance measurements were collected with an Analytical Spectral Devices (ASD) FieldSpec® 4 hyperspectral spectroradiometer (Malvern Panalytical, USA) between 350 and 2500 nm. The sensor was equipped with a light-diffusing fore optic remote cosine receptor (RCR) to measure planar irradiance. We selected three different locations and collected

spectral measurements for two samples each of green, red, and mixed snow algae patches, and two algae-free or "clean" snow areas, for a total of 24 measurement sites (2 of each of the 4 types across the 3 sites). Overall coastal snowpack in this region and at this time of year are generally comprised of dense wet snow, although snow density was not recorded. Snow density in Eq. 1 is based on average observations from similar study sites in the AP region in January 2020. Areas with snowmelt ponding were avoided. The RCR was placed upward to collect the downwelling planar irradiance incident upon

the snow surface ($E_d$) and the upwelling planar irradiance reflected from the snow ($E_u$). Measurements were collected in triplicate. The operator was located in a direction 90 - 135º away from the sun to minimize solar glint and self-shadowing (Mobley, 1999). Snow conditions did not allow for a tripod, so nadir orientation was determined by practice with a level and by visual assistance of an observer. Since the measurements were carried out under heavily overcast conditions where irradiance is dominated by the diffuse insolation with no solar azimuthal dependence, the influence of slight tilt when

measuring the downwelling irradiance (i.e. the cosine error) is expected to be minor (<0.5%) (Castagna et al. 2019). The reflectance measurements were taken prior to excavation of snow sample for laboratory analysis.

Post-processing of the data involved computing spectral reflectance, $R(\lambda)$, as the ratio of the upwelling flux normalized to the downwelling flux for each wavelength $(\lambda)$. The mean of the three measurements was calculated for each site. Ambient light conditions were too low in the short-wave infrared wavelengths for getting adequate signal-to-noise for

our measurements. In post-processing, reflectance values were truncated at 1350 nm for this analysis. This value represents the limit often used for RF calculations in other studies (Bryant et al., 2013). In addition, empirical correction coefficients were used to correct for temperature related radiometric inter-channel steps using the procedure and MATLAB code from Hueni et al. (2017). This removed the step function near 1000 nm for most of the spectra, although not fully for all spectra. However, this discontinuity does not significantly impact results or albedo calculations. Albedo was calculated as the

integrated $R$ in two different intervals: visible (400-700 nm) and infrared (700-1300 nm). Quality assurance was conducted for each spectrum and one station (Fildes 5, mixed algae) was removed from the data analysis because the spectral magnitude of $R$ was considered unrealistically high with the highest infrared albedo measured in the study (i.e. higher than clean snow) that was not consistent with the spectral shape and pigment content compared to the expected values at the other stations. For the remote sensing analyses, $R(\lambda)$ was convolved with the published spectral response functions (SRF) for the

Sentinel-2 sensor to obtain an estimate of bands that would be observed by the sensor after proper atmospheric correction centered at wavelengths in each part of the spectrum: blue (Band 2, 492 nm), green (Band 3, 560 nm), red (Band 4, 664 nm)





and near infrared (Band 5, 704 nm). Following Painter et al. (2001) and Gray et al. (2020), a scaled integral of Sentinel 2's Band 4 relative to a continuum formed by the line between Bands 3 and 5 (Rcont), $I_{B4}$ was calculated following:


$$I_{B4} = \frac{R_{ContB4} - R_{B4}}{R_{ContB4}} \qquad\qquad Eq. 2$$

### 2.3 Estimate of Pigment Absorption

The absorptance, $A(\lambda)$, attributed to each snow sample can be calculated from the reflectance as the amount of light that was absorbed (i.e., not reflected) from the snow where:


$$A(\lambda) = [1 - R(\lambda)] \qquad\qquad Eq. 3$$

To estimate the photosynthetic absorbptance attributed to algae, $A_{alg}(\lambda)$, we account for relative absorption compared to clean snow and remove the absorption due to effects of non-pigmented algal constituents and other factors such as water and impurities. We subtracted the absorptance measured at 709 nm where chlorophyll and accessory pigments have low absorption 205 following methods in bio-optical studies of seagrass and other vegetation (Zimmerman, 2003).

$$A_{alg}(\lambda) = [1 - R_{alg}(\lambda)] - [1 - R_{alg}(709)] \qquad\qquad Eq. 4$$

### 2.4 Albedo and Radiative Forcing

Albedo, α, is a measure of the diffuse reflectance of solar radiation from the snow surface that was estimated over two different wavelength regions for all the sample sites following:

$$\alpha = \frac{1}{N}\sum_{\lambda 1}^{\lambda 2} R(\lambda)\,\Delta\lambda \qquad\qquad Eq. 5$$

Where N is the number of wavelengths and calculated as the integral over two different wavelength regions visible (400-700 nm), $\alpha_{vis}$ and near infrared (700-1300 nm), $\alpha_{nir.}$


Net RF is calculated as the net flux upward (down minus up) and is calculated as:

$$F = E_d - E_u = E_d(1 - R) \qquad\qquad Eq. 6$$



This term depends on the amount of radiation reaching the snow surface, which is a function of time of year, time of day, as well as amount of cloud cover. Here, we follow previous studies to estimate the additional RF caused by the addition of algae compared to clean snow (Myhre et al., 2013) and replace the "1" with the reflectance of clean snow. Two different analyses of RF calculating instantaneous and interannual RF. We calculated Instantaneous RF as:

$$IRF \approx \sum_{350}^{850} E_d(\lambda) \left( R_{clean}(\lambda) - R_{algae}(\lambda) \right) \Delta\lambda \qquad Eq.\ 7$$

For comparability to other studies using clear-sky remote sensing imagery, we calculated IRF for clear sky solar irradiance normal to insolation at each site, ~62°S 58°W, on the date each site was visited in January 2018 at 13:00 local solar mean time using https://www.pvlighthouse.com.au (Ganey et al., 2017).


Because of the variability and prevalence of cloud cover in this region, we also conducted long-term flux analysis to determine what the forcing would be using realistic cloud cover for the region. Following from Bryant et al. (2013), a daily RF, $\bar{F}$, was estimated to provide the general magnitude of the forcing that might be expected given realistic cloud forcing and typical concentrations of snow algae. Hourly measurements of spectral irradiance were made at Palmer Station, Antarctica (64°46' S,

64°03' W, 21 m above sea level) with a SUV-100 spectroradiometer from Biospherical Inc. (Booth et al. 1995) processed between 400-600 nm from 1990-1997 (Dierssen et al., 2000). Irradiance was extrapolated to Photosynthetically Available Radiation (PAR) (400-700 nm) with a site-specific relationship using the radiative transfer model SBDart (Gautier and Landsfeld, 1997) run with different clouds, atmospheric and albedo conditions following:

$$E_d(PAR) = 1.42E_d(400-600) - 1.15 \qquad Eq.\ 8$$

Measurements of $E_d$ were also compared to clear sky formulations from the radiative transfer model with different surface albedo properties from 100% snow, 100% ocean and 50% snow/50% ocean. Modelled estimates of daily clear sky PAR are ~14% higher estimated with snow compared to ocean. Since snow and sea ice are generally absent from Palmer Station during

summer months, the radiative transfer modelling of clear sky irradiance was best approximated with an effective albedo of 100% ocean during these months (Fig. 3) and used in the analysis. Since algae are found on coastal snow and glacial margins during summer months, using a 100% snow albedo for radiative transfer simulations may overestimate the clear sky flux.





The formulation does not have exact spectral weighting between albedo and irradiance because the daily insolation was already
calculated for integrated photosynthetically available radiation (PAR) following from (Dierssen et al., 2000). The flux was
estimated using average albedos and daily irradiance following:

$$\bar{F} \approx E_{d,vis}\left(\alpha_{vis,clean} - \alpha_{vis,algae}\right) \qquad\qquad Eq.\ 9$$

Where $E_{d,vis}$ (W m$^{-2}$) is the daily average irradiance integrated across visible wavelengths (400-700 nm). Algal albedo $a_{vis,algae}$
were estimated as the individual means for the green and red algae, respectively across all sites and $a_{vis,clean}$ is the mean for the
clean snow sites.

### *2.5 Statistical Analyses*

Statistical tests were conducted in MATLAB®. Arithmetic means were calculated and shown with plus or minus the standard
deviation, unless otherwise indicated. A one-way repeated measures ANOVA was used to evaluate the impact of different
types of snow on the albedo with a significance level of 0.05. The central mark on the boxplot indicates the median, and the
bottom and top edges of the box indicate the 25th and 75th percentiles, respectively. The whiskers extend to the most extreme
data points not considered outliers, and the outliers are plotted individually using the '+' symbol. Model II simple linear
regression analyses were conducted to account for uncertainty in both the x and y variables.

## 3. Results and Discussion

### *3.1 Ground-based Spectral Albedo Measurements*

Spectral reflectance varied across all sites in a consistent manner with snow and algae composition (Fig. 4A-D). The reflectance
of visibly clean snow was greater than 0.80 in visible wavelengths and decreased into the near infrared (NIR) wavelengths
typical of snow spectra (Ganey et al., 2017). For most of the clean snow spectra, the reflectance was spectrally flat across the
visible wavelengths consistent with "white" snow. Several of the stations at Collins Bay, however, exhibited increasing
reflectance across blue and green wavelengths possibly due to the presence of impurities like dust (Mauro et al., 2015) or other
detrital matter. For all of the sites, the addition of light-absorbing algal pigments decreased reflectance in visible wavelengths.
The spectral reflectance of green snow algae was consistently lower than red algae and mixed algae fell in between the two
types. Albedo integrated across visible wavelengths (400 – 700 nm) ranged from 0.31 at a green snow algae site on Nelson
Island to 0.87 from a clean snow site at Collins Bay on King George Island (Table 1, Fig. 4E). The lowest (average ± standard
deviation) albedo in the visible wavelengths was for green snow algae patches (0.44 ± 0.12), followed by mixed snow algae





patches (0.58 ± 0.064), red snow algae patches (0.65 ± 0.09) and the highest albedo was at the clean snow sites (0.85 ± 0.043). A one-way between-subjects ANOVA conducted to compare the effect of snow algae on visible albedo found a significant effect at the $p<0.05$ level for the four conditions [$F_{3,19}$ = 23.68, p = $1 \times 10^{-6}$].

280         In contrast to visible wavelengths, no impact of algae on near infrared wavelengths was observed. All of the spectra tended to be very similar in near infrared wavelengths from 700-1300 nm with local minima related to water absorption features with the exception of two stations. The very low albedos in the NIR observed in the green snow algae patches on Nelson Island may have been influenced by the underlying ground due to a thin snowpack at these locations (7 and 8 cm, Table 1). The optical signal from the dark underlying ground likely contributed to lower values in the NIR wavelengths (~0.29). With the

exception of these two optically shallow stations, the NIR albedo from 700-1300 nm were similar across the different snow types such that the mean NIR albedo calculated from 700-1300 nm ranged only from 0.45 - 0.46 (Fig. 4F, Table 1). A one-way between subjects ANOVA conducted to compare the effect of snow algae on NIR albedo was not significant at the $p<0.05$ level for the four conditions [$F_{3,17}$ = 0.26, p = 0.86]. This result is consistent with light absorption by photosynthetic pigments influencing the visible or photosynthetically available radiation (PAR) and having no impact on the NIR albedo. The 15%

reduction in NIR albedo due to thin snow at the Nelson sites, however, illustrates the challenges in remote sensing LAPs in snow, such as BC compared to thin snow (Warren, 2013).

        The 20% reduction of visible albedo by red snow algae observed in this Antarctic study is roughly in the same range as Lutz et al. (2016), who reported a 13% albedo reduction caused by red snow algae in the Arctic. However, measurements in this study were collected from coastal Antarctic snow packs, whereas the study in the Arctic was conducted on glaciers,

which could lead to overall albedo differences due to dissimilarities physical snow conditions. Our spectral measurements were most similar to Ganey et al. (2017) who show a similar decrease in reflectance in visible wavelengths and minimal impact in NIR wavelengths. In contrast, Gray et al. (2020) show a marked difference in visible and NIR reflectance for two measurements of HDRF shown in their Fig. 2 but less so in the spectra in Supplemental Fig. 2. Painter et al. (2001) found a slight decrease in albedo in the NIR that was not observed for the thick snow samples assessed here. Differences in NIR

reflectance will be impacted by snow thickness and liquid water content of snow, but our results show no significant difference due to the presence of snow algae itself.

### 3.2 Pigment Concentration and Light Absorption

The concentration of pigments followed a large range across sites from less than detection limits in clean snow to nearly 100 mg ChP m$^{-2}$ found in a green snow algae patch on Nelson Island (Table 1). Consistent with past studies on Antarctic snow

algae (Davey et al., 2019a), pigment concentrations were overall much higher in green patches compared to red ones. Our chlorophyll $a$ plus phaeopigment concentrations ranged from 0.001 to 0.03 mg ChP m$^{-2}$ in visibly "clean" snow, 0.12 to 7.82 mg ChP m$^{-2}$ in red snow patches, 4.22 to 14.38 mg ChP m$^{-2}$ in mixed snow algae, and 8.82 to 141.78 mg ChP m$^{-2}$ in green snow algae patches (Table 1).



The estimated *Absorptance* (*A*) by snow algae reveal distinct patterns of broad absorption in soret or blue (400-500 nm) and a secondary peak in red wavelengths (678 nm) common to all algal groups (Fig. 5A). For these groups, green and mixed algae have higher magnitude of *A* than red algae due to enhanced pigment concentrations in green algae. The spectral shape of *A* normalized at 440 nm reveals that absorption by red algae is higher in the green wavelengths (500-570 nm) compared to green algae (Fig. 5B). This spectral difference is due to the enhanced concentrations of ancillary carotenoid

pigment Astaxanthin (Davey et al., 2019a) which absorbs in far blue and into green wavelengths of light, leading to a red color. Enhanced absorption due to carotenoids at 550 nm provide a means to discriminate between red and green algae using the reflectance spectrum (see below). When absorption is normalized to pigment concentration, the red algae absorb considerably more per mg of ChP compared to mixed or green algae (Fig. 5C). This difference is partly due to red algae having less chlorophyll *a* compared to carotenoids (Davey et al., 2019a). When normalized by an estimate of both ChP and ancillary

pigment (red dotted line Fig. 5C), pigment-specific *A* is three times lower but still higher than that for green algae. Differences in algal light absorption can also arise from differences in light adaptation, size, packaging within the cell and other factors (Kirk, 1991). A more thorough study on algal light utilization in relationship to these factors in different types of snow algae is warranted.

### 3.3 Radiative Forcing

The impact of snow algae on RF is dependent on a variety of environmental factors including the location, seasonality, daylength, duration of algal bloom, and the atmospheric conditions including cloudiness of the region. These sites, for example, were all measured under cloud-covered conditions that were prevalent throughout the sampling period. The instantaneous RF (IRF) produced under clear skies was greater than 200 W m$^{-2}$ at high algal concentrations (Fig. 6A). The average IRF for Green communities is more than double red communities (Table 1).

These results from Antarctica are similar to the only other estimates we could find of IRF, by red snow algae on an Alaskan icefield (Ganey et al., 2017), but here the relationship between IRF and pigment concentration was found to be logarithmic (Fig. 6B), explaining 70% of the variability in IRF. Although the role of microbes in surface darkening has been documented on the Greenland Ice Sheet, e.g. (Stibal et al., 2017), the associated RF has not been explicitly reported (Skiles et al., 2018). To our knowledge, Ganey et al. (2017) is the only other study to document the RF due to snow algae. Thus, our

results are some of the first studies to document the RF by snow algae in Antarctica and using realistic environmental conditions in the region. Clouds are extremely prevalent in this region of the Antarctic and clear-sky IRFs do not accurately represent the daily RF caused by snow algae. Following a previous study on dust (Bryant et al., 2013), mean daily RF was calculated based on hourly measurements of PAR at Palmer Station (Dierssen et al., 2000; Fig. 4C). Since algae had little impact on NIR albedo, the use of PAR (400-700 nm) is considered suitable for this analysis. As shown, daily PAR varied

considerably from clear sky conditions with the austral spring tending to be clearer relative to austral Fall (Fig. 6C). The





average RF of green and red snow algae varied throughout the course of the growing season with the mean RF by green algae close to 20 W m$^{-2}$ with a maximum up to 60 W m$^{-2}$ (Fig. 6D). Mean daily RF was highest around Ordinal Day 300 and declining after Day 350. Gray et al., (2020) estimated a 122-day growing season. Here we assume a 118-day growing season from Ordinal Day 352 (December 18$^{th}$ in non-leap years) to Ordinal Day 105 (April 15$^{th}$ in non-leap years) and found a mean RF is

26 W m$^{-2}$ for green algae and 13 W m$^{-2}$ for red algae.  However, maximums approach 40 W m$^{-2}$ for green algae and 20 W m$^{-2}$ for red algae at the height of the summer (Fig. 6E). Bryant et al. (2013) found that annual forcing from dust in snow ranged interannually from 20 to 80 W m$^{-2}$. Further, a snowmelt radiative transfer modelling study that simulated the dust-influenced snow cover evolution found that a daily mean average RF of 30 W m$^2$ advanced seasonal snowmelt by 30 days (Skiles and Painter, 2019). Hence, our mean daily RF results of 26 W m$^{-2}$ during the austral summer growing season for green algae

suggest snow algae in the polar regions is on par with that from dust in more temperate regions and has the potential to drastically alter snowmelt and regional hydrology, warranting further investigation.

The growing season for snow algae has not been well documented in this region. However, the requirement for liquid water for growth suggests the prevalence of snow algae may be greater in late summer and early austral Fall as the region warms. As shown, the radiative impact will be less in this time of the year (Ordinal Day 0-60) due to declining daylength and

relative increase in cloud cover. However, snow algae observations have also been shown to increase under cloudy skies vs days with intense sunlight due to a decrease in the degree to which the algae associate with the water-surface, thereby increasing the number of algae being removed from the top layer of snow by melt-water runoff during days with intense solar radiation (Grinde, 1983). Therefore, the timing and duration of red and green snow algae presence and their impact on RF in this region of Antarctica should be further explored. Lastly, previous results of low concentrations of BC in snow in the AP region (Khan

et al., 2019) and elsewhere on the continent suggest BC is not a driver of RF in the region outside of the proximity of field stations.

### 3.4 Bio-optics and Remote Sensing of Snow Algae

Many different heritage algorithms have been proposed for assessing algal pigment concentration from the reflectance spectrum. Our data show that light absorption by the snow algae is highly correlated to the logarithm of pigment concentration

(Fig. 7A). Similar to ocean microalgae (Dierssen and Randolph 2013), the relationships follow a logarithmic function with pigment concentration. As shown in Fig. 7A, the amount of blue reflectance (Sentinel Band 2 490 nm) explains 85% of the variability of the logarithm of ChP. For remote sensing purposes, using the absolute magnitude of reflectance is problematic due to issues with atmospheric correction and impurities. The amount of blue absorption relative to green (R560-R490) is also a similarly good predictor of ChP pigment and is more robust to such issues whereby the difference between Band 4 (560 nm)

and Band 3 (490 nm) explains 83% of the variability in pigment (Fig. 7B). Following Painter et al. (2001) and Gray et al. (2020), we also relate our pigment concentration to a scaled integral of Sentinel Band 4 (Eq. 2). Linear regression of the scaled integral of Sentinel 2's Band 4 (664 nm) relative to Bands 3 (560 nm) and 5 (704 nm) provides predictive power to estimate





higher concentrations of ChP ($>0.5$ mg m$^{-2}$) and can be applied when $I_{B4}$ is $>0$ (Fig. 7C). However, the relationship is not consistent across the full range of data shown. Gray et al. (2020) also note that red algae can alter the derived relationship

compared to green algae.

Differentiation of green and red algae can be optically achieved by exploiting the differences in green and red band reflectance. Differencing between Sentinel Bands 3 and 4 using red and green algae spectra shows smaller values compared to green algae or clean snow. Figures 7D and 7E illustrate how the difference in these three populations (red, green and clean snow) can be separated with the bands in Sentinel 2. However, we note this study lacks spectra from low concentrations of

green algae where the red and green algae populations would overlap.

Due to persistent cloud cover in this region, we were unable to obtain suitable imagery concurring with our sampling time. While we have focused on Sentinel-2 because of its high spatial and temporal resolution and heritage approaches for remote sensing of snow algae, these approaches could be easily converted to other sensors with different spatial and spectral resolutions. The approaches using only blue, green, and red bands are robust and could be applied to other sensors like Landsat-

OLI and Planetscope. Additional spectral capabilities from sensors like Worldview-2 and -3 and hyperspectral sensors could provide the opportunity to further quantify ancillary pigments and other ecological parameters of the algae.

## 4. Potential Regional Climate Impacts

Snow algae are highly dependent on their habitat conditions (e.g. radiation, water availability, and temperature) and are generally dormant over winter, coming to life as snow begins to melt because the water brings nutrients, often from guano

deposits within the snow. Therefore, it is likely that snow algae growing season is responsive to climatic changes. As demonstrated here and previous studies, snow algae reduce surface albedo by absorbing more solar radiation than the surrounding snow and ice. This potential 'bioalbedo' feedback (Cook et al., 2017) is not currently accounted for in global climate models. Furthermore, it could be significant in polar climates where marginal snow pack areas are extensive and retreating (Benning et al., 2014; Hodson et al., 2017; Tedesco et al., 2016).

Finally, we present a cursory calculation of the overall RF by red and green snow algae in the AP region by taking the Gray et al. (2020) green snow algae surface area estimate of 1.9 km$^2$ for the northern AP and our average daily RF calculation of 26 Wm$^{-2}$ by green snow algae and 13 W m$^{-2}$ for red snow algae.  This results in a total daily RF by green snow algae of 50 MJ and 24 MJ by red algae. Although the timing and duration of snow algae blooms are not well documented, if we assume a 118-day growing season from December 18$^{th}$ to April 15$^{th}$, this translates to a seasonal increase in energy absorbed

at the snow surface of 51 x 10$^7$ MJ season$^{-1}$ by green snow algae and 25 x 10$^7$ MJ season$^{-1}$ by red snow algae over the northern AP study region. Furthermore, if 334,000 J are needed to melt 1 kg of snow (Cohen, 1994), this results in roughly 1.53 Mt or 2522 m$^3$ of snow melted by green-colored algae and 7.37 Mt or 1218 m$^3$ of snow melted by red-colored algae, a serious impact on snow in the algae-covered regions versus adjacent areas. Compared to the total annual regional melt It should be noted that we think the spatial and temporal distribution of green and red snow algae are not equal throughout the growing season and



should also be further investigated. As the climate warms, the spatiotemporal distribution of red and green algae along the Northern AP will change, likely growing in latitudinal and landward extent, increasing the RF effect. Lastly, the impacts of RF by snow algae on snowmelt in the AP region are not currently accounted for in estimates of Antarctic ice-free habitat expansion, e.g. (Huss and Farinotti, 2014; Lee et al., 2017). Our calculations above suggest snow algae play a significant role in snowmelt in the AP regions where they occur, which could be amplified in the future as the climate warms.

**5. Future Outlook**

This study builds on past studies, providing additional insight into the spectral differences in Antarctic snow impacted by red and green algae, as well as presents some of the first RF estimates of red and green coastal Antarctic snow algae using realistic solar forcing. The cloudiness of the region makes it extremely challenging to accurately model in terms of radiative transfer and from satellites that require clear skies. We show, however, that even with cloud cover, the mean daily RF in visible

wavelengths due to the presence of snow algae is on par with that published for dust impacts in Colorado and much greater than BC impacts in the Arctic (Painter et al., 2012; Skiles et al., 2018; Skiles and Painter, 2019). Further, the average IRF and mean daily RF for green communities is more than double red communities. These values suggest the impact on long-term RF in polar ecosystems through altered snowmelt and regional hydrology, as well as the need to map and monitor red and green snow algae communities.

We also demonstrate the potential to map the spatial and temporal distributions of red and green snow algae using optical signatures of enhanced green absorption by the red algae compared to the green. Future studies could further explore the range of green and red algal pigment concentrations and applicability of these approaches to satellite imagery. Eventually, these mapping algorithms may be used to track expansion of algal bloom extent and pigment content and determine other climate- and ecosystem- factors controlling algal growth. The ultimate benefit of using satellite imagery will be the ability to

monitor spatial and temporal changes across larger areas of the cryosphere. The expansion of these techniques to estimate ancillary pigment concentrations will also be possible with future hyperspectral sensors, e.g., NASA's Plankton Aerosol and ocean Ecosystem (PACE) and Surface Biology and Geology (SBG) missions.

The AP experienced significant warming in the last several decades of the 20[th] century, with the hottest day on record (18.4°C) recently recorded at Esperanza Station on 2020-02-06 in the AP region (Robinson et al., 2020). The warming trend

and persistently warmer conditions in the AP could be resulting in more snow-algae growth in coastal areas, which in turn could have a positive feedback on surface albedo and melting. The potential feedback from the RF of red and green snow algae outlined here should be accounted for in global climate models in order to properly account for snowmelt and ice edge retreat, which could be significant in this region of Antarctica, especially as the climate continues to warm. Additionally, it is likely that the snow algae bloom season is being extended due to warming temperatures. These data may eventually be used for

algorithms that could help facilitate an understanding of the spatial and temporal variability of red and green snow algae in this changing region. Furthermore, snow algae contribute to net primary productivity (NPP) and carbon biogeochemistry.





Further research can explore methods relating pigment concentrations with NPP of snow algae. This study contributes to the understanding of this natural phenomenon in the cryosphere and the feedbacks to radiative forcing in the AP.

**Acknowledgements.** ALK was supported by a Fulbright Scholarship with the Chilean Antarctic program. ALK and TS were supported by a Landsat Science Team award to T. Scambos, USGS Contract # 140G0118C0005. HMD acknowledges funding from NASA Ocean Biology and Biogeochemistry PACE Science Team;  RRC acknowledges the support of the Consejo Nacional de Ciencia y Tecnología (CONICYT, Preis Fondecyt 1191932 & 1171690) and the Chilean Antarctic Institute (INACH, Preis RT32-15 & RT70-18). JH was supported by CONICYT [FONDAP-IDEAL 15150003] and FONDECYT
[POSTDOCTORADO 3180152]. The authors appreciate help from Edgadro Sepulveda and Nicole Torres, as well as preliminary discussions on this work with Dr. Pete Convey at the British Antarctic Survey and Dr. Matthew Davey at Cambridge University. All data for this publication are included in this paper and archiving of the spectral albedo data is currently underway for input into the endmember database being put together by NASA: https://speclib.jpl.nasa.gov. The data are included as supplementary information while the archiving step is completed.


**Author contributions.** A.L.K. conceived of the study, collected the snow samples.  A.L.K and H.M.D analyzed the spectral albedo and pigment data and conducted the absorptance, and radiative forcing analyses.  H.M.D provided the Palmer Station irradiance data. A.L.K, T.S. and R.C. contributed to the study design. J.H analyzed the chlorophyll samples. A.L.K, T.S. and H.M.D. contributed to the data analysis, algorithm development and interpretation. All authors contributed to writing and
editing of the manuscript.

**Competing Interests.** There are no conflicts of interest.

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



**Table 1. Pigment concentration, albedo, and RF calculated for each snow type sampled with corresponding metadata.**

| Site[a] | Chl (µg/L) | ChP (mg/m²) | IRF (Watts /m²) | $\alpha_{vis}$ | $\alpha_{nir}$ | Ord. Day | Latitude | Longitude | Elevation (m) | Snow Depth (cm) |
|---|---|---|---|---|---|---|---|---|---|---|
| **Clean Snow** | | | | | | | | | | |
| F1 | 0.02 | 0.003 | 24.70 | 0.8000 | 0.4144 | 15 | -62°12'24.7" | 58°57'53.1" | 11 | 75 |
| F2 | 0.06 | 0.009 | 4.89 | 0.8378 | 0.4561 | 15 | -62°12'25.1" | 58°57'53.3" | 3 | 70 |
| F3 | 0.01 | 0.005 | 0.00 | 0.8457 | 0.4774 | 15 | -62°12'25.1" | 58°57'52.8" | 0 | 60 |
| C7 | 0.17 | 0.024 | 0.00 | 0.9119 | 0.4894 | 18 | -62°10'02.4" | 58°51'17.1" | N.A. | 43 |
| C8 | 0.17 | 0.024 | 9.31 | 0.8915 | 0.4813 | 18 | -62°10'02.4" | 58°51'17.1" | N.A. | 45 |
| N1 | 0.01 | <0.001 | 22.77 | 0.7974 | 0.3887 | 20 | -62°15'48.5" | 58°56'47.7" | 17 | >300 |
| N2 | 0.01 | <0.001 | 0.00 | 0.8475 | 0.4058 | 20 | -62°15'47.6" | 58°56'48.4" | 12 | >300 |
| *Mean* | *0.064* | *0.013* | *8.810* | *0.847* | *0.445* | | | | | |
| **Red Algae** | | | | | | | | | | |
| F4 | 7.39 | 1.377 | 45.38 | 0.7378 | 0.4510 | 15 | -62°12'25.3" | 58°57'52.4" | 1 | 40 |
| F8 | 1.06 | 0.115 | 41.17 | 0.7376 | 0.4980 | 15 | -62°12'24.8" | 58°57'52.0" | 0 | 20 |
| C2 | 43.73 | 7.187 | 185.60 | 0.4868 | 0.4048 | 18 | -62°10'02.4" | 58°51'17.1" | N.A. | 30 |
| C6 | 35.45 | 5.974 | 110.51 | 0.6436 | 0.4703 | 18 | -62°10'02.4" | 58°51'17.1" | N.A. | 40 |
| N3 | 6.70 | 0.879 | 79.47 | 0.6394 | 0.4245 | 20 | -62°15'44.7" | 58°56'49.1" | 1 | 30 |
| N4 | 11.34 | 1.319 | 66.04 | 0.6648 | 0.4804 | 20 | -62°15'44.6" | 58°56'48.9" | 2 | 23 |
| *Mean* | *17.61* | *2.81* | *88.0* | *0.652* | *0.455* | | | | | |
| **Green Algae** | | | | | | | | | | |
| F6 | 32.11 | 8.816 | 64.86 | 0.6804 | 0.4792 | 15 | -62°12'25.2" | 58°57'52.4" | 2 | 30 |
| F7 | 215.70 | 40.828 | 154.06 | 0.4657 | 0.4698 | 15 | -62°12'25.1" | 58°57'52.2" | 3 | 32 |
| C1 | 369.82 | 71.387 | 199.85 | 0.4298 | 0.4461 | 18 | -62°10'02.4" | 58°51'17.1" | N.A. | 30 |
| C4 | 222.36 | 43.134 | 219.20 | 0.3952 | 0.3943 | 18 | -62°10'02.4" | 58°51'17.1" | N.A. | 28 |
| N6[a] | 672.00 | 99.230 | 227.53 | 0.3181 | *0.2992[a]* | 20 | -62°15'45.2" | 58°56'51.2" | 4 | **7[a]** |
| N7[a] | 158.78 | 13.977 | 212.63 | 0.3594 | *0.2920[a]* | 20 | -62°15'45.2" | 58°56'51.2" | 4 | **8[a]** |
| *Mean* | *278.5* | *46.23* | *179.7* | *0.441* | *0.447* | | | | | |
| **Mixed Algae** | | | | | | | | | | |
| *F5[b]* | *31.43* | *4.216* | *5.47* | *0.8159[b]* | *0.5243[b]* | *15* | *-62°12'25.3"* | *58°57'52.4"* | *2* | *30* |
| F9 | 15.55 | 3.795 | 118.89 | 0.5598 | 0.4513 | 15 | -62°12'24.7" | 58°57'51.7" | 1 | 27 |
| C3 | 23.69 | 5.065 | 119.14 | 0.6175 | 0.4814 | 18 | -62°10'02.4" | 58°51'17.1" | N.A. | 37 |
| C5 | 74.07 | 14.381 | 161.78 | 0.5028 | 0.4813 | 18 | -62°10'02.4" | 58°51'17.1" | N.A. | 30 |
| N5 | 37.90 | 4.571 | 75.63 | 0.6474 | 0.4386 | 20 | -62°15'45.0" | 58°56'50.3" | 4 | 20 |
| *Mean* | *37.80* | *6.95* | *118.86* | *0.58* | *0.46* | | | | | |

Abbreviations: F=Fildes; C=Collins; N=Nelson; Chl=Chlorophyll *a* in melted snow; ChP = Sum of Chlorophyll *a* and

Phaeopigments per snow surface area; IRF = Instantaneous RF assuming a clear sky; $\alpha_{vis\ vis}$ = Albedo in visible (400-700 nm);



$\alpha_{nir}$ = Albedo in near infrared (700-1300 nm); Ord. Day= Ordinal Day in 2018; Elevation was not available (N.A.) for Collins site.

[a]Due to the shallow snow depth (7-8 cm), the albedo is likely influenced by dark underlying surface compared to optically deep snow and these stations were excluded from NIR albedo calculations where algal pigments do not significantly absorb.

[b]This station had unrealistically high albedo given its type and spectral shape and was not included in the RF and spectral

analyses in the paper.

**Figure 1: Map of sample locations and photos of sampling sites. Map images are from Landsat 8, acquired 29 September 2014, Path 207, rows 103 and 104.**






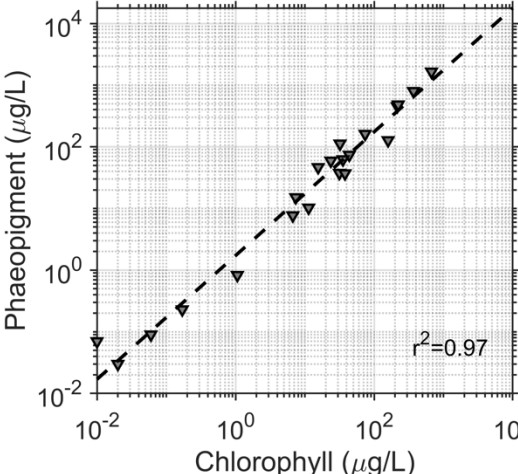


**Figure 2: Correlation *between* Chlorophyll and Phaeopigments for all samples (R²=0.97) with a slope of 2.34. Concentrations were added together for the subsequent analysis.**


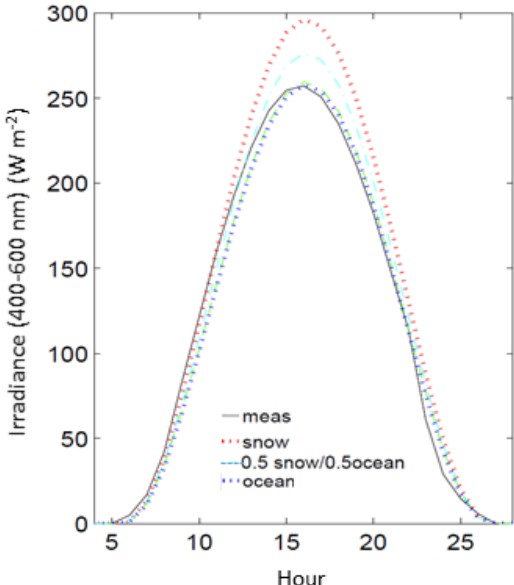

**Figure 3. Measured Irradiance (meas) for an example day at Palmer Station (Ordinal Day 300, 1990) compared to models of clear sky irradiance using different surface albedos (snow, ocean and a 50% snow/ocean mixture).**




**Figure 4. Spectral Reflectance colored according to each type of snow algae measured at each location from Fig. 1 (A-C) and for all the stations together (D). Albedo was calculated following Eq. 9 for each snow type for E) visible wavelengths (400-700 nm) and F) near infrared wavelengths (700-1300 nm).**






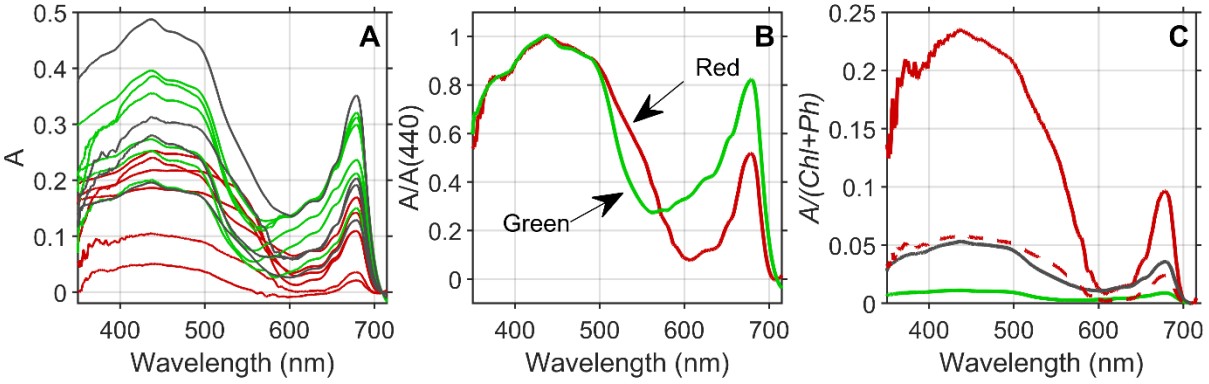

**Figure 5. A) Estimated Absorptance** *(A)* **of the snow algae reveal distinct patterns of broad absorption in soret or blue (400-500 nm) and a secondary peak in red wavelengths (678 nm) common to all algal groups, with green (green lines) and mixed (black lines) algae presenting higher absorption magnitude than red algae (red lines) due to enhanced pigment concentrations. B) The spectral shape of mean absorption normalized at 440 nm reveals that absorption by red algae (red line) is higher in the green wavelength compared to green algae (green line), which provide a means to discriminate between red and green algae using the reflectance spectrum. C) Absorption normalized to chlorophyll plus phaeopigment biomass demonstrates that red algae (red line) communities absorb considerably more per mg of pigment compared to mixed (black line) or green (green line) algae patches. The dotted red line includes estimated pigment contribution from astaxanthin esters in red algae (3:1 relative to Chlorophyll a; Davey et al. 2019).**


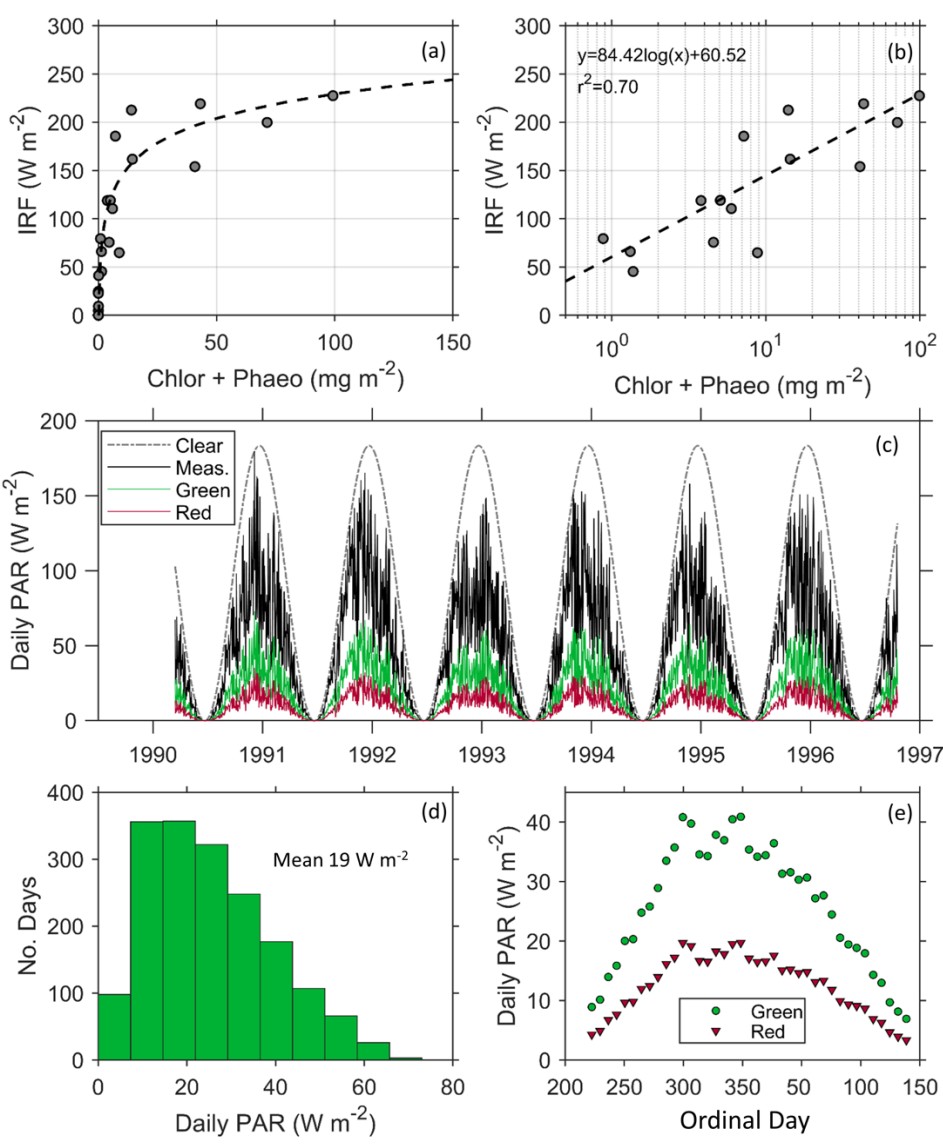

**Figure 6. A) Instantaneous radiative forcing (IRF) under clear skies increases with increasing Chlorophyll a and**
**phaeopigment concentrations and B) is modelled using a linear relationship with the logarithmic concentration of**
**pigment. C) Daily PAR measured at Palmer Station (Dierssen et al. 2000) indicates the prevalence of clouds compared**
**to clear sky (dotted line) and provides a daily average radiative forcing (RF) due to red and green algae. D) A histogram**
**showing the daily forcing of green algae follows a slightly skewed pattern with a mean of 19 W m⁻². E) The daily RF of**
**green and red algae averaged weekly over the course of the growing season at Palmer Station.**


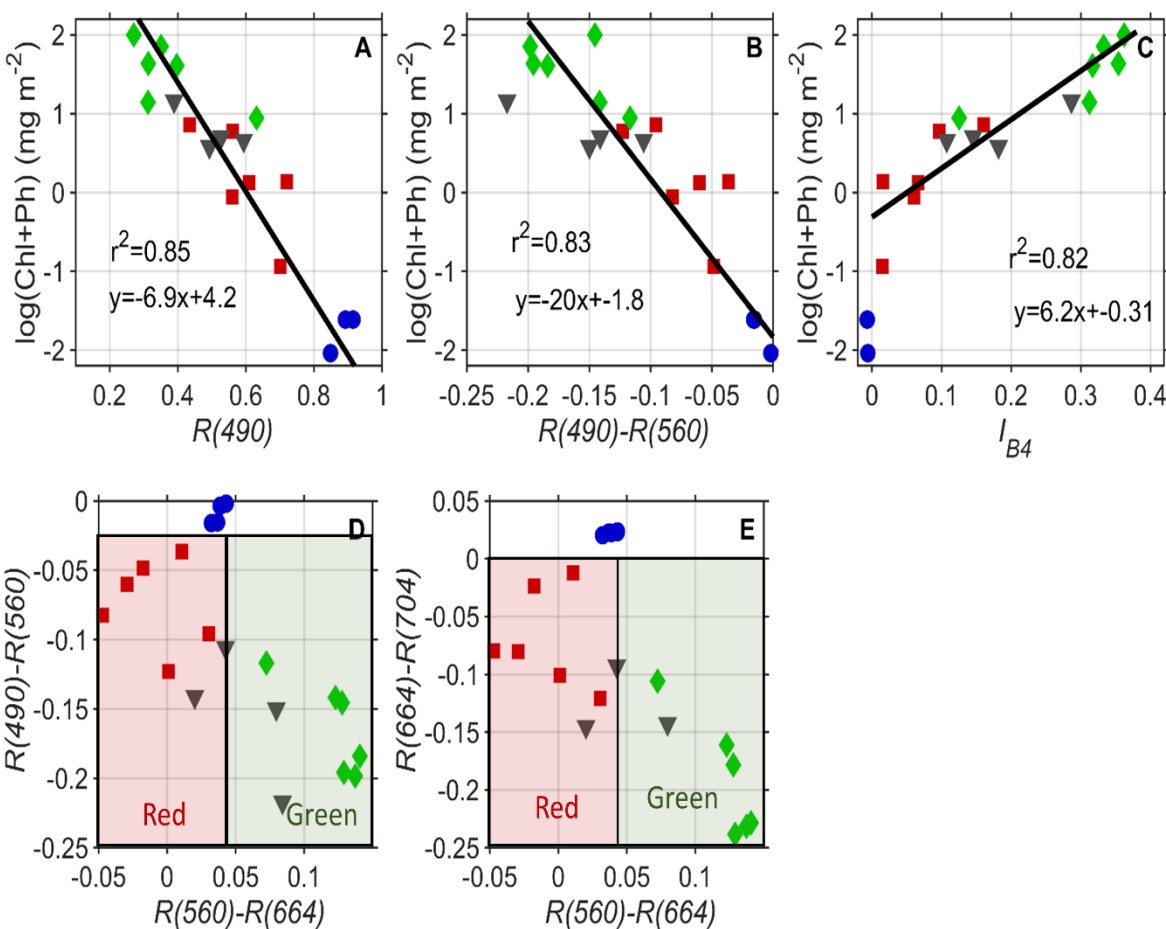

**Figure 7. A) The reflectance in blue wavelengths (490 nm) is inversely correlated to the logarithmic concentration of pigment. B) The difference in blue to green bands from Sentinel 2 explain 83% of the variability in pigment and can be used for remote sensing of pigment. C) The scaled integral in Sentinel 2 Band 4 (660 nm) is related to pigment concentrations for indices >0. Red and green algae can be differentiated easily based on their differential absorption in green wavelengths 560 nm compared to D) red and blue bands and E) red and infrared bands. Red squares are red algae; green diamonds are green algae; black triangles are mixtures of the two; blue circles are clean snow.**