# Peer review of "Spectral Characterization, Radiative Forcing, and Pigment Content of Coastal Antarctic Snow Algae: Approaches to Spectrally Discriminate Red and Green Communities and Their Impact on Snowmelt"

_The Cryosphere, 2020_

## Referee Comment (RC1) · Anonymous Referee #1 · 1 Sep 2020

General Comments:

This seems to be a good, descriptive story about two color morphs (red and green) of snow algae on Antarctic Peninsula islands; the role that each morph plays in radiative transfer to the snow there; and how the two can be distinguished using remote sensing products. The authors do an effective job of describing how they calculated radiative forcing due to algae and found it similar to another estimate a hemisphere away on

a maritime glacier in Alaska (Ganey et al., 2017). This latter result may not be too surprising, considering that the latitudes (60 N and 62 S) have similar solar insolation and the algal reflectance spectra, too, look alike. The Antarctic context with two colors of snow algae that appear rather straightforward to separate using remote sensing is quite interesting. I wonder how much of the difference in Figure 4 is due to abundance and how much is due to actual pigment differences? I wonder how much of Figure 6 is a bit too speculative.

Overall, I look forward to seeing revisions.

Specific Comments:

Which leads to the following questions for the authors that this reviewer would appreciate in a revision.

1) Was there any measure of algal abundance? Cell counts for instance? Sort of interesting that green algal blooms absorbed so much more than the red ones. Is it due to algal abundance or pigments? For example, those curves in Figure 4 look like other papers' figures where what varies is the abundance of algae, rather than color. This could be addressed to satisfy the curiosity of some readers.

2) Would the authors be willing to discuss this caption from Figure 5 "C) Absorption normalized to chlorophyll plus phaeopigment biomass demonstrates that red algae (red line) communities absorb considerably more per mg of pigment compared to mixed (black line) or green (green line) algae patches" in the context of the Dial, Ganey, Skiles 2018 hypothesis that natural selection favors the red-colored pigment in red-colored snow algae to melt more snow, thereby freeing-up water and nutrients needed for life (in lines 315-317, perhaps).

3) Might it be worth reviewing some of the key differences between biotic and abiotic LAP as described in the opening paragraph of Ganey et al. 2017 to highlight the fact that living organisms act on the cryosphere in a fundamentally different way than

mineral particles or black carbon do?

4) Line 38+: "This intense warming is likely increasing snowmelt availability, potentially impacting red and green snow algae blooms, which are sensitive to light (Rivas et al., 2016)" could include other citations including:

The original paper claiming that snow algae are sensitive to light and that pigments protect.

Bidigare R, Ondrusek ME, Kennicutt MC et al. Evidence a photoprotective for secondary carotenoids of snow algae. J Phycol 1993; 29:427–34.

A paper suggesting that perhaps photoprotective pigments have another role.

Gorton HL, Williams WE, Vogelmann TC. The light environment and cellular optics of the snow alga Chlamydomonas nivalis Bauer Wille. J Photochem Photobiol 2001; 73:611–20.

A theoretical and experimental paper that algal pigments are not just protective, but functional in melting snow and ice.

Dial RJ, Ganey GQ, Skiles SM. What color should glacier algae be? An ecological role for red carbon in the cryosphere. FEMS Microbiology 2018; 93.

5) Lines 325-329: Not clear if these are for Fig 6 C or something else. Fig 6 C needs a bit more fleshing out both in the methods and in the results. If I'm reading between the lines, this is using algae spectra from the late two-thousand-teens and measured PAR from the late nineteen-nineties to show a possible time course of radiative forcing due to various colors/mixtures of algae? And what about figure 6e? One year recently or an average of the Palmer data in the 1990s? These results seem speculative at best. Snow alga abundance is notoriously variable from year to year, so not sure assuming this constant balance of red vs green year after year is informative. Perhaps avoid what could be construed as over-interpretive modeling….or maybe I'm missing the point. What is/are the point/s?

Technical Corrections:

Line 46, 51, 53, etc.: misplaced comma? misplaced parentheses? Using the "eg.," I'd think that abbreviation would be inside the parens....or is this a standard practice for the journal (doesn't appear to be; suggest dropping all the "eg." Followed by a citation?

Line 47: "influenced by inputs from penguin and seal excreta, which helps to fertilize local glacial, terrestrial, and aquatic ecosystems (Hodson, 2006)." Plural subject implies use of plural verb: "help" rather than singular form?

Line 61: Recent paper also pertinent: Onuma, Y., Takeuchi, N., Tanaka, S., Nagatsuka, N., Niwano, M. and Aoki, T., 2020. Physically based model of the contribution of red snow algal cells to temporal changes in albedo in northwest Greenland. The Cryosphere, 14(6), pp.2087-2101. Also might be worth pointing out some other qualitative differences between biotic and abiotic LAPs as mentioned in Ganey et al.

Lines 64-78 could be reduced to a simple statement that the taxonomy is unstable.

Line 82: algae is a plural word: "Snow algae were first mapped..." would be correct.

Line 85: Replace as "On the Greenland ice sheet"?

Line 94: "are clean (free of snow algae)" does this mean clean of all LAP? Or just algae? Maybe say more simply "are free of snow algae"?

Line 98: "several heritage approaches to estimate pigment concentration" what's a "heritage approach"?

Line 105: "with slightly less frequent wildlife traffic." Than what/where?

Methods: are all, some, or no sites glacial? Hard to infer except that the snowpacks are optically thick, but my impression is that none are glacial: correct? Although Figure 1 suggests the possibility of glacial edge?

The references need major revision to clean up and make useable to readers.

[Figure]

Some examples of incomplete citations (there may be more):

Ganey, G. Q., Loso, M. G., Burgess, A. B. and Dial, R. J.: forcing on an Alaskan icefield, , (September), 490 doi:10.1038/NGEO3027, 2017

Mobley, C. D.: Estimation of the remote-sensing reflectance from above-surface measurements, 1999.

Painter, T. H., Duval, B., Thomas, W. H., Heintzelman, S., Dozier, J. and Mendez, M.: Detection and Quantification of Snow Algae with an Airborne Imaging Spectrometer Detection and Quantification of Snow Algae with an Airborne Imaging Spectrometer, , 67(11), doi:10.1128/AEM.67.11.5267, 2001.

Takeuchi, N., Dial, R., Kohshima, S., Segawa, T. and Uetake, J.: Spatial distribution and abundance of red snow algae on the Harding Icefield , Alaska derived from a satellite image, , 33, 1–6, doi:10.1029/2006GL027819, 2006.

---

## Referee Comment (RC2) · Yangyang Liu (Referee) · 17 Sep 2020

[referee-annotated manuscript omitted]

---

## Author Comment (AC1) · 15 Oct 2020

The author's appreciate the constructive comments from Reviewer 1. Please find our revisions below.

1) Was there any measure of algal abundance? Cell counts for instance? Sort of interesting that green algal blooms absorbed so much more than the red ones. Is it due to algal abundance or pigments? For example, those curves in Figure 4 look like

[Figure]

other papers' figures where what varies is the abundance of algae, rather than color. This could be addressed to satisfy the curiosity of some readers.

Author's Response 1) The authors agree cell counts would be a useful addition and plan to include cell counts or a measure of algal abundance in subsequent sampling campaigns. However, we did not have a way to preserve the samples for cell counts during this study, so we cannot address the question as to whether the reduction of albedo is higher for green patches due to pigments or abundance alone. For this study, only pigments (Chl a and Phaeopigments) were analyzed for a measure of algal biomass. However, based on our correlation of IRF to Chlor + Phaeo (r2 = 0.7; Fig. 6B), it appears the pigments play a role in the absorption of solar radiation. We have amended the paper in Methods where we state "Cell counts were not conducted in this study." In the results we have added a statement "Future work will better elucidate whether the reduction of albedo is higher for green patches due to pigment concentrations or cell abundance."

2) Would the authors be willing to discuss this caption from Figure 5 "C) Absorption normalized to chlorophyll plus phaeopigment biomass demonstrates that red algae (red line) communities absorb considerably more per mg of pigment compared to mixed (black line) or green (green line) algae patches" in the context of the Dial, Ganey, Skiles 2018 hypothesis that natural selection favors the red-colored pigment in red-colored snow algae to melt more snow, thereby freeing-up water and nutrients needed for life (in lines 315-317, perhaps).

Author's Response 2) Yes, the following text has been added near lines 315 – 317, "The red color of astaxanthin has also been shown to play an adaptive role in melting snow and ice in order to make water and nutrients available for algal growth (Dial et al., 2018)."

3) Might it be worth reviewing some of the key differences between biotic and abiotic LAP as described in the opening paragraph of Ganey et al. 2017 to highlight the

fact that living organisms act on the cryosphere in a fundamentally different way than mineral particles or black carbon do?

Author's Response 3) The following third paragraph has been amended as follows, "Both biotic and abiotic light absorbing particles (LAPs) (Skiles et al., 2018) lead to a reduction in surface albedo. LAPs are generally comprised of dust (e.g. Bryant et al., 2013; Painter et al., 2012; Skiles and Painter, 2019), black carbon e.g.,(Khan et al., 2019; Rowe et al., 2019), volcanic ashes (Flanner et al., 2007) and snow algae (e.g. Ganey et al., 2017; Lutz et al., 2016). LAPs influence spectral albedo in the visible spectrum, 400 – 700 nm (Warren and Wiscombe, 1980). Spectral albedo is further dependent on physical snow properties such as specific surface area, i.e., grain size and shape (e.g. Cordero et al., 2014), liquid water content, surface roughness, snow depth, albedo of underlying ground (for thin snow packs), and snow density (Flanner et al., 2007). Aged snow that has collected LAPs, typically has an albedo around 0.5 – 0.7, and in extreme cases of organic LAP content, can range below 0.2 (Khan et al., 2017). Arctic red snow algae blooms can reduce surface albedo up to 13% (Lutz et al., 2016). Snow darkening by LAPs and the associated radiative forcing (RF) have the potential to impact the long-term climate, while accelerating snow melt and changes in regional hydrology in the near term. Given their anthropogenic association they are not expected to diminish in the future (Skiles et al., 2018). Furthermore, living organisms such as snow algae reduce surface albedo differently than LAPs, due to their life-cycle response to light (Bidigare et al., 1993; Rivas et al., 2016), the functional role of their pigments in melting snow and ice (Dial et al., 2018), as well as their strong absorption features as opposed to LAPs broad absorption across the visible wavelengths. However, unlike other LAPs (Flanner et al., 2007), the 'bioalbedo' feedback (Cook et al., 2017; Onuma et al., 2020) from microbes living and growing on the surface of the cryosphere is not currently accounted for in global climate models and only one study has assessed the RF of red snow algae, in Alaska (Ganey et al., 2017).

4) Line 38+: "This intense warming is likely increasing snowmelt availability, potentially
impacting red and green snow algae blooms, which are sensitive to light (Rivas et al., 2016)" could include other citations including: The original paper claiming that snow algae are sensitive to light and that pigments protect. Bidigare R, Ondrusek ME, Kennicutt MC et al. Evidence a photoprotective for secondary carotenoids of snow algae. J Phycol 1993; 29:427–34. A paper suggesting that perhaps photoprotective pigments have another role. Gorton HL, Williams WE, Vogelmann TC. The light environment and cellular optics of the snow alga Chlamydomonas nivalis Bauer Wille. J Photochem Photobiol 2001; 73:611–20. A theoretical and experimental paper that algal pigments are not just protective, but functional in melting snow and ice. Dial RJ, Ganey GQ, Skiles SM. What color should glacier algae be? An ecological role for red carbon in the cryosphere. FEMS Microbiology 2018; 93.

Author's Response 4) These citations have been added to Line 38+ and the reference list.

5) Lines 325-329: Not clear if these are for Fig 6 C or something else. Fig 6 C needs a bit more fleshing out both in the methods and in the results. If I'm reading between the lines, this is using algae spectra from the late two-thousand-teens and measured PAR from the late nineteen-nineties to show a possible time course of radiative forcing due to various colors/mixtures of algae? And what about figure 6e? One year recently or an average of the Palmer data in the 1990s? These results seem speculative at best. Snow alga abundance is notoriously variable from year to year, so not sure assuming this constant balance of red vs green year after year is informative. Perhaps avoid what could be construed as over-interpretive modeling....or maybe I'm missing the point. What is/are the point/s?

Author's Response 5) Figures 6C-E have to do with the amount of light and actual radiative forcing of red and green algae based on their albedo, and do not refer to algal abundance of red and green snow algae. Since the Antarctic Peninsula region is very cloudy, a clear sky albedo estimate would overestimate the actual radiative forcing. Thus, the aim of the figure is to convey the radiative specific to any patch of red or

green snow algae. We have added text to explain this better in the beginning of Section 3.3 "The impact of snow algae on RF is dependent on a variety of environmental factors including the location, seasonality, daylength, duration of algal bloom, and the atmospheric conditions including cloudiness of the region. Here, we evaluate radiative forcing in two ways: 1) using clear sky instantaneous estimates of solar radiation and 2) using long-term measurements of daily radiative forcing at this location. The latter approach is particularly important because it represents a more conservative and realistic treatment of the impact of algae in this region based on the local cloud climatology. Importantly, these results are not dependent on the specific areal extent of each type of algae, but represent the radiative forcing per patch of algae detected in any location in the region and can be scaled to the average radiation available during a particular time in the growing season."

We have also amended the legend of Figure 6c to, "C) Modelled influence of a patch of red and green snow algae on the seasonal radiative forcing (RF) using daily PAR measured at Palmer Station from 1990 – 1997 (Dierssen et al. 2000) and the mean measured surface spectral albedo for each algal type. Measured PAR (solid line) is significantly lower than modelled clear sky PAR (dotted line) and provides a more realistic estimate of RF"

Technical Corrections: Line 46, 51, 53, etc.: misplaced comma? misplaced parentheses? Using the "eg.," I'd think that abbreviation would be inside the parens. . ..or is this a standard practice for the journal (doesn't appear to be; suggest dropping all the "eg." Followed by a citation?

Author's Response: These have been amended to, (e.g. reference).

Line 47: "influenced by inputs from penguin and seal excreta, which helps to fertilize local glacial, terrestrial, and aquatic ecosystems (Hodson, 2006)." Plural subject implies use of plural verb: "help" rather than singular form?

Author's Response: This has been amended to the singular form.

Line 61: Recent paper also pertinent: Onuma, Y., Takeuchi, N., Tanaka, S., Nagatsuka, N., Niwano, M. and Aoki, T., 2020. Physically based model of the contribution of red snow algal cells to temporal changes in albedo in northwest Greenland. The Cryosphere, 14(6), pp.2087-2101. Also might be worth pointing out some other qualitative differences between biotic and abiotic LAPs as mentioned in Ganey et al.

Author's Response: The Onuma reference has been added to this line. The following text has also been added near Line 61, "Furthermore, living organisms such as snow algae reduce surface albedo differently than LAPs, due to their life-cycle response to light (Bidigare et al., 1993; Rivas et al., 2016), the functional role of their pigments in melting snow and ice (Dial et al., 2018), as well as their strong absorption features as opposed to LAPs broad absorption across the visible wavelengths."

Lines 64-78 could be reduced to a simple statement that the taxonomy is unstable.

Author's Response: This paragraph has been shortened/amended to, "In our study region, the snow algae are comprised of green algae (Chlorophyta), but the taxonomy are unstable. Based on findings on nearby Adelaide Island they are likely a combination of Chloromonas, Chlamydomonas, and Chlorella genuses (Davey et al., 2019). In this study, we did not have the opportunity to analyze the community composition of our samples, but plan to include this in future work."

Line 82: algae is a plural word: "Snow algae were first mapped. . ." would be correct.

Author's Response: Thank you, this change has been made.

Line 85: Replace as "On the Greenland ice sheet"?

Author's Response: This change has been made.

Line 94: "are clean (free of snow algae)" does this mean clean of all LAP? Or just algae? Maybe say more simply "are free of snow algae"?

Author's Response: This change has been made.

[Figure]

Line 98: "several heritage approaches to estimate pigment concentration" what's a "heritage approach"?

Author's Response: The word heritage has been removed.

Line 105: "with slightly less frequent wildlife traffic." Than what/where?

Author's Response: This sentence has been amended to, "It is approximately 200 meters above mean high tide, with slightly less frequent wildlife traffic than the other two sites."

Methods: are all, some, or no sites glacial? Hard to infer except that the snowpacks are optically thick, but my impression is that none are glacial: correct? Although Figure 1 suggests the possibility of glacial edge?

Author's Response: The following sentences have been added to the end of the first paragraph of the methods, "Only the algae-free site at Nelson was on a glacier, the rest were coastal snow packs. Both Nelson and Collins were near glacier edges."

The references need major revision to clean up and make useable to readers. Some examples of incomplete citations (there may be more): Ganey, G. Q., Loso, M. G., Burgess, A. B. and Dial, R. J.: forcing on an Alaskan icefield, , (September), 490 doi:10.1038/NGEO3027, 2017 Mobley, C. D.: Estimation of the remote-sensing reflectance from above-surface measurements, 1999. Painter, T. H., Duval, B., Thomas, W. H., Heintzelman, S., Dozier, J. and Mendez, M.: Detection and Quantification of Snow Algae with an Airborne Imaging Spectrometer Detection and Quantification of Snow Algae with an Airborne Imaging Spectrometer, , 67(11), doi:10.1128/AEM.67.11.5267, 2001. Takeuchi, N., Dial, R., Kohshima, S., Segawa, T. and Uetake, J.: Spatial distribution and abundance of red snow algae on the Harding Icefield , Alaska derived from a satellite image, , 33, 1–6, doi:10.1029/2006GL027819, 2006.

Author's Response: Thank you so much for catching these. I have been having some

issues with my citation manager, Mendeley and see that these were not correctly imported into the bibliography of the manuscript. These have hopefully all been addressed now. Thanks again for catching this.

---

## Author Comment (AC2) · 15 Oct 2020

[revised manuscript text omitted]